# Variables Influencing per Capita Production, Separate Collection, and Costs of Municipal Solid Waste in the Apulia Region (Italy): An Experience of Deep Learning

**DOI:** 10.3390/ijerph18020752

**Published:** 2021-01-17

**Authors:** Fabrizio Fasano, Anna Sabrina Addante, Barbara Valenzano, Giovanni Scannicchio

**Affiliations:** 1Regione Puglia—Sezione Ciclo Rifiuti e Bonifiche, 70125 Bari, Italy; addante.sabrina@gmail.com (A.S.A.); g.scannicchio@regione.puglia.it (G.S.); 2Regione Puglia—Dipartimento Mobilità, Qualità Urbana, Opere Pubbliche e Paesaggio, 70125 Bari, Italy; b.valenzano@regione.puglia.it

**Keywords:** municipal solid waste, deep learning, municipal collection centers, door to door service, separate collection, waste management

## Abstract

Municipal solid waste (MSW) must be managed to reduce its impact on environmental matrices and population health as much as possible. In particular, the variables that influence the production, separate waste collection, and costs of MSW must be understood. Although many studies have shown that such factors are specific to an area, the awareness of these factors has created opportunities to implement operations to enable more effective and efficient MSW management services, and to specifically respond to the variables that have the most impact. The deep learning approaches used in this study are effective in achieving this goal and can be used in any other territorial context to ensure that the organizations that deal with these issues are more aware and create useful plans to promote the circular economy. Our findings indicate the important influence of number of rooms in a residential buildings and construction years on MSW production, the combination of services such as municipal collection centers and door-to-door service for separate MSW collection and the characteristics of the residential buildings in the municipalities on MSW management costs.

## 1. Introduction

Waste management, in particular municipal solid waste (MSW), is one of the most important problems faced by modern society. As declared by The 7th Environment Action Programme (EAP), it is necessary to create a circular economy based both on the reduction MSW production and to increase reuse and/or recycling as much as possible [1,2]. Separate collection is one of the most important instruments that can be used to reduce residual waste streams and the landfilling of waste [3], and, consequently, land consumption and landfill gas emissions [4]. Source separation can lead to regional economic benefits using locally recycled materials instead of imported raw materials [5].

Many studies showed that many factors (for example, geographical location, collection repetition, standard of living, economic condition, laws on waste management, local culture and beliefs, population growth, and size of households) affect the production, efficiency of separate collection, and management costs of waste [6,7,8]. As many factors as possible should be considered in a multivariate analysis approach to understand not only the factors influencing MSW but also the degree of importance of each factor. Multivariate analysis on MSW management in some studies [6,7,8,9,10,11,12,13] was performed with a machine learning approach, which are computer algorithms that improve automatically through experience. Machine learning algorithms build a mathematical model based on sample data, known as training data, to make predictions or decisions without being explicitly programmed to do so [14]. The availability of large data sets and their processing using graphics processing units (GPUs) have promoted the development of new modeling approaches such as deep learning, which is an evolution of machine learning approach that enabled the construction of nets with many hidden layers. This resulted in a new theory and caught the attention of many researchers [15,16,17].

The innovation of this study is in our approach multivariate analysis with this new type of model and our consideration of a large number varied factors (102 factors) that could influence per capita production, separate collection, and cost of MSWs (infrastructure, organizational, demographic, social, and economic factors including organizational and economic services of treatment, waste disposal, and separate collection). The aim was to understand in depth the objects of study as well as to plan interventions for the most influential factors that favor the circular economy, reduce MSW production and the associated management costs, and increase separate collection. Apulia Region—Waste Management Section and Apulian Waste Regional Observatory (AWRO) regulations were made also these studies [18]. AWRO is a technical-administrative structure that aims to monitoring the Apulian integrated waste cycle and to support the Apulian region in terms of its environmental policy on waste management by collecting and elaborating data on urban solid waste and special/hazardous waste production, recovery, and disposal, publishing the results.

## 2. Background of Multivariate Analysis on MSW Management in Europe and in Italy

Methods dealing with only one variable are called univariate methods. Methods dealing with more than one variable at once are called multivariate methods. Natural systems cannot be described satisfactorily using univariate methods because any particular phenomenon studied in detail usually depends on several factors. If these factors are collected every day a multivariate data matrix is generated. For interpretation of such data sets multivariate data analysis is useful. Multivariate data analysis can be used to process information in a meaningful fashion. These methods can afford hidden data structures. On the one hand the elements of measurements often do not contribute to the relevant property and on the other hand hidden phenomena are unwittingly recorded. Multivariate data analysis allows us to handle huge data sets in order to discover such hidden data structures which contributes to a better understanding and easier interpretation. There are many multivariate data analysis techniques available [19] ranging from the classic inferential statistical method regression model (Poisson) [20,21,22] to new models based ANN machine learning algorithms as deep learning or random forest [16,23,24]. Most of the studies on MSW management in Europe are concentrated on the cities areas and used every type of methods to predict MSW generation. MSW generation was predicted using regression and trend analysis in Romania [25] for example. In this case population aged 15–59 years strongly influences the results. A regression model was also used in Turkey where it resulted that GDP has a high impact on the MSW generation as it is directly related to consumption [13]. An ANN model was used, for example, in Serbia to determine future waste characteristics [26], showing an interesting relation between MSW and average incomes of municipality, level of employment, age structure, educational level and housing condition. Other studies developed a general regression neural network (GRNN) model for the prediction of annual municipal solid waste (MSW) generation at the national level for 44 countries of different size, population and economic development level [9]. In the Italian context there are not many papers about multivariate analysis on MSW management. An interesting paper studied to what extent income and municipal waste generation are linked and at what level of income they become delinked using data sets from Italian provinces that include rich northern and poorer southern regions [27].

## 3. Deep Learning Theory

Since the 1950s, a small subset of artificial intelligence (AI) methods, called machine learning (ML), has revolutionized several fields in the last few decades. Neural networks (NN) in turn is a subfield of ML, and it was this subfield that spawned deep learning (DL). Since its inception DL has been creating ever larger disruptions, showing outstanding success in almost every application domain. DL which uses either deep architectures of learning or hierarchical learning approaches), is a class of ML developed largely from 2006 onward. Learning is a procedure consisting of estimating the model parameters so that the learned model (algorithm) can perform a specific task. For example, in artificial neural networks (ANN), the parameters are the weight matrices. The recent literature states that DL-based representation learning involves a hierarchy of features or concepts, where the high-level concepts can be defined from the low-level ones and low-level concepts can be defined from high-level ones. In some articles, DL has been described as a universal learning approach that is able to solve almost all kinds of problems in different application domains [16,28,29,30,31].

The DL approach is sometimes called universal learning because it can be applied to almost any application domain. It is a robust approach because it does not require precisely designed features. Instead, optimal features are automatically learned for the task at hand. As a result, the robustness to natural variations of the input data is achieved. The same DL approach can be used in different applications or with different data types. This approach is often called transfer learning. In addition, this approach is helpful where the problem does not have sufficient available data. In addition, The DL approach is highly scalable [16,28,29,30,31].

Deep Learning approach can be supervised, semi-supervised, and unsupervised. Supervised learning is a learning technique that uses labeled data. In the case of supervised DL approaches, the environment has a set of inputs and corresponding outputs. Semi-supervised learning is learning that occurs based on partially labeled datasets. Unsupervised learning systems are ones that can without the presence of data labels. Deep learning approaches can be used to solve forecasting problem of classification or regression [16,28,29,30,31]. In particular deep-learning structures for nonlinear classification and regression are the most frequently used for modeling and forecasting [32] and was used in this study. Neurons, which are placed in the layers, are the basic processing elements of deep learning models. The layers between the first layer (inputs) and the last layer (outputs) are called hidden layers. Neurons on each layer sum the weighted inputs, add a bias to the sum, and then apply an activation function to process the sum and compute the outputs. The signal processing of neurons can be mathematically expressed as:$$Y_{i}=\int \left(\overset{M}{\underset{j=1}{\sum }}\left(W_{ij}X_{j}+\beta_{ij}\right)\right)$$
where *Y_i_* is the output to the ith neuron in the current layer;$\;w_{ij}\;$and $\beta_{ij}$ are the weight and bias of the *j*-th input on the *i*-th neuron, respectively; *M* is the number of inputs;$\;X_{j}$ is the *j*-th output from the previous layer; and *f* is the activation function, which is in this study, is the most frequently used [16,17], called rectified linear input (ReLU; Figure 1).

## 4. Integrated Waste Management Theory

In Italian and European law [35], integrated management is a system aimed at managing the entire waste process (including production, collection, transport, treatment, and final destination) with the aim of energy and raw materials recovery, and, therefore, to minimize the fraction destined to the landfill, and whose activities, even the realization and management of the plants, are entrusted to a single subject. The law addresses the issue of waste by outlining a number of priorities and actions within the integrated problem management logic. Specifically, the priorities are: development of clean technologies, design and placing on the market of products that do not contribute or make a minimum contribution to the generation of waste and pollution, technological improvements to eliminate the presence of hazardous substances in waste, and active role of public administrations in the recycling of waste and its use as an energy source. The actions mention:

**Waste prevention**:Correct assessment of the environmental impact of each product during its entire life cycle;Procurement specifications considering the ability to prevent production;Promote experimental agreements and programs to prevent and reduce the quantity and danger of waste;Implement integrated pollution reduction and prevention actions.

**Waste recovery**:Reuse and recycling;Production of secondary raw material by treating the waste;Favoring the market in re-used products via economic measures and specifications in tenders;Use of waste to produce energy (energy recovery (cold biological oxidation, gasification, and incineration).

Therefore, if the first level of attention is focused on the need to prevent the formation of waste and reduce its dangerousness, the transition to a circular economy shifts the focus to the reuse, fixing, renewal, and recycling of materials and products existing in Italy. What is normally considered waste can be turned into a resource. Products are specifically tailored to material cycles: consequently, these create a flow that maintains added value for as long as possible. The residual waste is close to zero. A circular type project is the starting point for taking any new product or service in the circular economy. Considering the durability, reuse, repair, reconstruction, and recycling, cars, computers, appliances, packaging, and many other products can be designed. Greater cooperation within supply chains can reduce costs, waste and damage to the environment.

Advances in eco-innovation offer new products, processes, technologies, and organizational structures. Some companies have the opportunity to discover new markets by moving from selling products to selling services and developing business models based on renting, sharing, repairing, updating, or recycling individual components. This new approach yields many business opportunities for small- and medium-sized enterprises (SMEs).

Recycling is a fundamental step for the circular economy because it means transforming waste into a resource (raw material, substance, or product). To recycle, however, it is necessary to know when, under what conditions, and for what purpose waste ceases to be a resource (end of waste). However, the end of the waste cannot be decided by the recycler, but must be determined by the authority. To date, European or national standards have only been decided for glass, metals, waste fuels, ground asphalt, door-to-door service, and vulcanized rubber, which allow the transformation from waste to resource. For the non-reusable and then recycled material (such as paper napkins) and the sub-items (i.e., the indistinguishable and therefore non-recyclable fraction of waste), there are two solutions for energy recovery through cold or hot systems, such as biooxidation (aerobic or anaerobic), gasification, pyrolysis and incineration, or landfill disposal. Therefore, even in an ideal situation of complete recycling and recovery, a percentage of residual waste is disposed of in landfills or oxidized to eliminate it and recover energy. Ideally, for the Italian and European Law [35], use of incineration and unsorted landfills should be limited to the minimum necessary. Although landfilling proves to be the better financial option, it is for the shorter term. The landfill option would require the need of a replacement landfill much sooner [36]. Instead, recycling saves more energy than is generated by incinerating mixed solid waste in an energy-from-waste facility [37,38,39].

## 5. Materials and Methods

### 5.1. Study Design

The sequential steps used in the development of the study design of the three variables, Annual per capita waste production, MSW separate collection and MSW management costs were:(1)*Data Collection.* This step was about searching and collecting all the data available for develop Multivariate analysis. Annual data from 2008 to 2018 on each municipalities were collected. In this way, the analyses included the different socio-economic characteristics of producers and their evolution.(2)*Choice of possible influential factors to be analyzed.* From the data collected, in this step were identified factors for the study of the three dependents variables. Some independent factors can be excluded due to a lack of a possible influence on the analyzed dependent variable.(3)*Treatment of Descriptive Indicator and Missing Data.* Same factor collected can be descriptive. These factors must be converted into a numerical format to be applied to Multivariate analysis. The techniques used for this purpose was “ordinal coding” [40]. If the data were found to be inconsistent or untreatable, they were excluded from the analysis(4)*Development of Deep Learning Models.* To understand which variables most affected the dependent variables (per capita production of MSW, separate waste collection, and costs for waste collection and disposal), numerous models with a deep learning approach must be developed and tested, changing both the number of hidden layers and the number of neurons within these layers and learning epochs.The steps for the model standard development were:(a)Data were randomized.(b)Independent parameters were normalized to a single comparable unit of measurement (with values ranging from 0 to 1, using the following formula [41]:
Xn= (Xnn−Min(X))/Max(X)−Min(X)
where *Xn* is the normalized value of each variable for record *n*, *Xnn* is the non-normalized value of each variable for record *n*, *Max*(*X*) is the maximum value of each variable, and *Min*(*X*) is the minimum value of each variable.(c)All datasets must be divided into two parts, the first part containing 70% of the data to train the model, and the second, containing 30% of the data, were subsequently used to evaluate the quality of the model (testing phase) [42]. With the first part, the developed models adapt their algorithms to become increasingly precise on the basis of a series of learning epochs defined by the builder of the model.Each model is supervised and solves regression problems. For each layer of the models, a bias neuron was provided to strengthen its effectiveness. ReLU is used for each model [33,34].All models was developed using the R 3.6.3 program (Keras).(5)*Choice of the most effective model.* The accuracy of all developed models was evaluated by using the trained model on the testing data and comparing predicted values with actual testing values. Two measures were used to evaluate Model Quality: root mean square error (RMSE) and mean absolute error (MAE) [43]. Low values of each type of error correspond to better model performance. The R 3.6.3 program (Keras) was used to calculate these values.(6)*Identification of factors most influenced each of the three independent variables within each model.* To assess which factors most influenced each of the three independent variables within each model, was used the R package “vip” (permutation-based VI scores method) developed in 2018 [44]. The individual conditional expectation (ICE) curves were subsequently constructed to underline the relationship between the most important variables in the models and the output of the model. The ICE curves represent the output values resulting from the model for each value that can be taken from each dataset record (gray lines) [45]. The red lines of each figures represent average output values as the value of each variable changes. Individual conditional expectation (ICE) plots, is a tool for visualizing the model estimated by any supervised learning algorithm developed in 2015 [45]. From the variation in this red line, we determined the range of increase or decrease in the average output to increase of the unit of measure of each variable. In this calculation, the highest error of each model between RMSE and MAE was considered, using following equations:Maximum range value:*Max*(*X*0 ± (*Max*(*RMSE*: *MAE*))) − (*X*1 ± (*Max*(*RMSE*: *MAE*)))/|*Y*0 − *Y*1|Mean range value:(X0−X1)/|Y0−Y1|Minimum range value:*Min*(*X*0 ± (*Max*(*RMSE*: *MAE*))) − (*X*1 ± (*Max*(*RMSE*: *MAE*)))/|*Y*0 − *Y*1|
where *X*0 is the output value at v normalized input value of 0, *X*1 is the output value at a normalized input value of 1, *Y*0 is the non-normalized input value = 0, and *Y*1 is the non-normalized input value = 1.(7)*Inferential statistics analyses to confirm the results of the models.* Cronbach’s α, Pearson’s correlation coefficient, Shapiro–Wilk normality test, Kruskal–Wallis test, and Dunn’s post-hoc test) were conducted with R.3.6.3 software, considering a *p*-value < 0.05 as a statistically significant difference to confirm the results of the model.

### 5.2. Geographical Study Area

Apulia, which is a region of south-eastern Italy with 257 municipalities, a land surface area of 19,347 km^2^, 800 km of coastline and four million inhabitants was selected as the area for this study [20]. MSW management services are organized in 38 homogenous areas. Every area have an homogeneous territorial or population characteristics. For example there are areas characterized by hilly territory and with greater rural economic activity, while there are coastal and flat areas with greater commercial activities [46] (Figure 2).

Apulia has the highest MSW per capita production in southern Italy (467 kg/inhabitant/year vs. 499.7 kg/inhabitant/year in Italy in 2018) and is the fourth last region in terms of percentage of separate collection (47.3% vs. 58.1% in Italy in 2018) [48].

### 5.3. Details about Step 1—Data Collection

Data were collected from different sources of each municipality for the period 2008–2018:

#### 5.3.1. Population, Demographic Indicators, Land Area, Residential Buildings, Local Units of Active Businesses

Data were collected from http://dati.istat.it [49]. websites regarding: resident population (total, by age, by sex, by marital status); density (inhabitant per km^2^); number of components per family; demographic indicators (mortality rate, birth rate, natural balance, migration balances, total balance); land area; number of residential buildings by period of construction, by number of rooms, and by number of floors; and number of local units of active businesses by type.

#### 5.3.2. MSW Production and Separate Collection Data

MSW production data included waste type consisting of everyday items that are discarded by the public. Although the waste may originate from a number of sources that has nothing to do with a municipality, the traditional role of municipalities in collecting and managing these kinds of waste have produced the particular etymology “municipal”. These data are transmitted monthly by the Apulian municipalities on regional environmental portal [50] and are available.

The percentage of separate collection was calculated based on guidelines introduced by the Italian Environmental Decree of 26 May 2016 and implemented by the Apulia region [51]. These data, which are classified by European Waste Codes (ECW), are grouped by main fractions of separate collection (paper, plastic, glass, organic, multi-material, green, and metals).

#### 5.3.3. Tax Returns

Data regarding tax returns (number of taxpayers, and total income = EUR 0, 0 to 10,000, 10,000 to 15,000, 15,000 to 26,000, 26,000 to 55,000, 55,000 to 75,000, 75,000 to 120,000, and >120,000) is available on the website of the Italian Ministry of Finance [52].

#### 5.3.4. Land Use

Data about areas for industrial agriculture, traffic (road building) and especially urban human settlements percentage on the total land extension per Municipalities can be found on the ISPRA website [53].

#### 5.3.5. Municipalities’ Budget and MSW Management Services

The budget data concerning the expenses for the collection and disposal of waste were provided by the Italian Ministry of the Interior from 2008 to 2018. In particular about purchases of goods and services, personnel costs and investment expenditure. These data included information about MSW management services from 2008 to 2015 (type of administrative management of MSW collection, transportation and treatment services. For example managed directly by Municipalities or contracted to private company).

#### 5.3.6. Municipal Waste Collection Centers and Door-to-Door Collection

A survey was addressed to all Apulian municipalities, using the Google Forms (Google LLC, Mountain View, CA, USA). Data collected for each municipality by this survey included: presence of waste collection centers (residential drop-off facility for recycling and trash disposal)., kilograms of MSW separate fraction collected in waste collection centers from 2008 to 2018, Door-to-door services implementation date (During door to door collections, recyclable materials or food waste are collected directly from residents’ doorsteps in communal corridors) and MSW management services from 2016 to 2018. Before submitting the questionnaire to all municipalities, a sample of 15 municipalities was involved. Based on the replies, we calculated Cronbach’s α to measures reliability or the internal consistency of the survey [54]. The same test was subsequently applied to all questionnaire replies. The Cronbach’s α of the sample of 15 municipalities was 0.97. This indicator was subsequently calculated as 0.92 on all respondents (231/257, 89.9%).

#### 5.3.7. Tourist Arrivals and Presences

The Apulian data of tourist arrivals (number of tourists who visited the Apulia region) and tourist presence (tourist arrivals multiplied by the days of stay) were obtained through the website of the Regional Agency Puglia Promozione [55].

#### 5.3.8. Coastal or Rural Municipalities

Data from cartographic analysis were obtained using the QGIS 3.10 program (QGIS.org, 2020. QGIS Geographic Information System. QGIS Association. http://www.qgis.org, OSGeo, Chicago, IL, USA).

### 5.4. Details about Step 2—Choice of Possible Influential Factors to be Analyzed

From the data collected, we identified 102 factors for the study of the three dependents variables. Some independent factors were excluded due to a lack of a possible influence on the analyzed dependent variable.
(1)We chose 88 independent factors for the dependent variable “Annual per capita waste production (kg)”. These data were corrected considering number of tourists and tourist arrivals(2)We chose 99 independent factors for the dependent variable “Percentage of separate collection”.(3)We chose 101 independent factors for the dependent variable “MSW management costs (Euros per inhabitant per year)”.

### 5.5. Details about Step 3—Treatment of Descriptive Indicator and Missing Data

Of the 102 factors collected, 4 were descriptive. These factors were converted into a numerical format using the techniques of ordinal coding [40] as follows:MSW management Apulian homogenous areas: Each of the 38 homogeneous areas was separately assigned a code from 1 to 38;Coastal or rural municipalities: Coastal municipalities were coded 2; rural areas were coded 1;MSW management services: The same numerical classification used by the Italian Ministry of the Interior was used:(1)Service managed in economy (directly by the municipality)(2)Service managed with municipal company (by company controlled by the municipality)(3)Service managed with provincial company (by company controlled by provincial public authority)(4)Service managed with a consortium (created by the union of two or more municipalities but does not include the municipality in question)(5)Service managed with a private company(6)Service managed with a public company (but not controlled by the municipality)(7)Consortium management service, consortium head (created by the union of two or more municipalities, including the municipality in question as head of the consortium)(8)Consortium management service, consortium body (created by the union of two or more municipalities, including the municipality in question as body of the consortium)(9)Service with other type of managementDoor-to-door service: If present all year, a value of 1 was assigned; otherwise, 0. If services started during the course of the year, the numerical classification depended on the following formula:
Months of active door-to-door service in a year/Number of months in the year

1.7% of the data collected were found to be inconsistent or untreatable and were excluded from the analysis. Some studies asserted that a missing rate of 5% or less is inconsequential, others maintained that statistical analysis is likely to be biased when more than 10% of data are missing [56,57]. In any case, our study had a lower value of data inconsistent or untreatable.

## 6. Results

This section presents the results of the deep learning models chosen according to the methods described in Section 5 with respect to each of the three variables considered in the study.

### 6.1. Annual Per Capita Waste Production

Annual per capita waste production (kg inhabitant + tourists per year) in Apulia decreased by 24.3% from 2008 to 2018 from 516.9 to 467.2 kg per capita. The dataset with all complete input data contained 2730 records (96.6%, 2730/2827. This number is above the 2417 records that were needed to have a 99% confidence level and a 1% confidence interval). The best performing model had three hidden layers of 60, 20, and 10 neurons each (Figure 3).

The correlation between the real values in the testing dataset (30% of the data) and those predicted from the model was 94.6% (MAE = 30.8 kg and RMSE = 40.2 kg).

The most important variables influencing annual per capita waste production per the model (Figure 4, are mainly related to the rooms in a residential building (1 room—X29, 2 rooms—X30, 5–8 rooms—X32, 3–4 rooms—X31, and 9–15 rooms—X33), residential building construction years, and, in particular, to the frequency of the oldest residential buildings (buildings built <1918—X20, buildings built between 1919 and 1945—X21, and buildings built between 1946 and 1960—X22), the type of municipality (coastal or rural—X6), and the frequency of the declared incomes on the total of the declared incomes, and, in particular, to the lower bands of income (EUR 26,000–55,000—X67, 0–10,000—X64, 15,000–26,000—X66, and 10,000–15,000—X65).

Figure 5 shows the ICE curves of the 15 variables most influencing annual per capita waste production derived by the model.

Table 1 shows the range of the average increase/decrease per unit of measurement for each of the 15 variables shown in Figure 5 using the calculation method described in Section 5.1.—Step 6.

Figure 5 and Table 1 show that trend in the number of rooms variables in residential buildings was inversely proportional to the annual per capita waste production. The variable that had the greatest impact on annual per capita waste production was X29 (percentage of residential buildings with 1 room) showed a marked and linear downward trend in the ICE curve.

If the percentage of this type of building in all municipalities was equal to the minimum in the dataset (11.2% of total buildings), the average annual per capita waste production in Apulia would be almost double compared to the current (800 kg/year). With each percentage point increase in these buildings, annual per capita waste production is reduced by an average of 5.3 to 7.1 kg.

Only the buildings with more rooms (X33) showed a trend directly proportional with per capita waste production. With each percentage point increase in these buildings, annual waste production increased by an average range of 1 to 9.3 kg. A higher percentage of older residential buildings (X20, X21, and X22) seemed to favor a lower annual waste production. If all Apulian municipalities (X6) were coastal, the per capita waste production would be higher. The average of the variable is 80 kg (range from 0.4 to 160.4 kg). The income of the population influences MSW production (in particular, low- and medium-income classes). The increase in the percentages of these classes led to an increase in MSW production per capita.

### 6.2. Percentage of Separate Collection

The percentage of separate waste collection in Apulia has quadrupled from 12.4% in 2008 to 51.4% in 2019. The dataset with all complete input data contained 2623 records (92.8%, 2623/2827. This number is above the 2417 records needed to have a 99% confidence level and a 1% confidence interval). The best-performing model had four hidden layers with 90, 70, 20, and 10 neurons each (Figure 6).

The correlation between the real values in the testing dataset (30% of the data) and those predicted from the model was 94.8% (MAE = 2.9% and RMSE = 3.9%).

The most important variables influencing the percentage of separate waste collection per the model (Figure 7) are related to door-to-door service (X94) and the main waste fraction of separate collection. In order of importance, they are: organic fraction, paper, multimaterial fraction, and glass (X96, X99, X102, and X100, respectively). Annual MSW per capita production is important for the percentage of separate collection (X4). Among the most important variables and least discounted were some of the infrastructural features of residential buildings such as structures with larger rooms (>16 rooms (X34) and 9–15 rooms (X33)) or residential buildings with one floor (X35) and four floors (X38). Municipal collection centers (X95) and their percentage of collected waste compared to the total of separate collection were found to play an important role in influencing the separate collection (16th variable in order of importance, affecting 1.3% of the model, which is the same value obtained for X20 (percentage of residential buildings built <1918/total residential buildings).

Figure 8 shows the ICE curves of the 16 variables most influencing percentage of separate waste collection derived by the model.

Table 2 shows the range of the average increase/decrease per unit of measurement of each of the 16 variables shown in Figure 8 using calculation method described in Section 5.1.—Step 6.

Figure 8 and Table 2 show that door-to-door service led to an increase in the percentage of separate collection from 3.2% to 18.8%, with a mean of 11%. With the increasing percentage of separate collection, the percentage of organic fraction in total separate collection increased, while the other fractions tended to decrease proportionally (paper, multimaterial fraction, and glass).

Annual MSW production was found to have an inversely proportional effect on the percentage of separate collection. With each 1 kg decrease in waste production, the separate collection increased in the range of 0.011% to 0.022%. Separate collection tended to decrease with the increase in the size of residential buildings (over 16 rooms and over four floors).

Municipal waste collection centers influenced separate collection. For each increase in percentage of separate collection gathered by the municipal waste collection centers, separate collection increased in the range of 0.02% to 0.18%

An inferential statistical analysis was conducted to confirm the model results. In particular, the separate collection data were compared between the municipalities and the different type of MSW services. The distribution of the separate collection percentages of the available dataset was not normal (Shapiro–Wilk W-test = 0.87552, *p*-value < 0.0001). The nonparametric Kruskal–Wallis test was used for comparison (Kruskal–Wallis chi-squared = 1250.7, degree of freedom (df) = 3, *p*-value < 0.0001), showing a difference between the medians of the percentage of separate collection of the groups.

Dunn’s post-hoc test showed there was statistically significant differences for each comparison (all *p*-values < 0.0001 excluding municipalities with municipal waste collection centers (MWCC) vs. municipalities with neither of the two services (*p* = 0.004)). The percentage of separate collection was higher in the municipalities with door- to-door service (DtD) or with municipal waste collection centers (MWCC) and even more if they had both (MWCC+DtD) (Figure 9).

### 6.3. MSW Management Costs (EUR Per Inhabitant Per Year)

In 2018, the municipalities spent EUR 716,744,600 on MSW management in the Apulia region. Since 2008, expenditure has grown by 39.7%. Most costs in this area were used for the purchase of goods and services (over 96%)

The dataset with all complete input data included 2618 records (92.6%, 2618/2827. This number is above the 2417 records needed to have a 99% confidence level and a 1% confidence interval). The best performing model had three hidden layers of 80, 40, and 20 neurons separately (Figure 10).

The correlation between the real values in the training dataset (30% of the data) and those predicted from the model was 86.2% (MAE = EUR 24.2 and RMSE = EUR 32.1).

The most important variables influencing MSW management costs (Figure 11) were found to be related to the characteristics of the residential buildings in the municipalities X35 (residential buildings with one floor), X20 (residential buildings built <1918), X21 (residential buildings built from 1919 to 1945, X37 (residential buildings with three floors), X29 (residential buildings with one room), and X23 (residential buildings built from 1961 to 1970)). Among the top 10 variables were the type of municipality (coastal or rural (X6)), the presence of door-to-door service (X94), and tourist arrivals (X16). The divorced population (X43) also seemed to have an influence. Among the separate collection fractions collected, organic appeared to have the greatest influence (X96).

Figure 12 shows the ICE curves of the 15 most influential variables affecting MSW management costs derived by the model.

Table 3 shows the range of the average increase/decrease per unit of measurement of each of the 15 variables shown in Figure 12 using the calculation method described in Section 5.1.—Step 6.

Figure 12 and Table 3 show that municipal costs lowered as residential buildings with one floor and older than 1918 increased. The coastal municipalities, as a model, seem to have a slight tendency to have higher costs. The increase in one percentage point of the divorced population in relation to the total population increased in costs from EUR 10.73 to 44.52.

For every increase in tourists arriving in the Apulian municipalities, costs increased from EUR 0.0003 to 0.00005. For each increase in one unit per km^2^ of catering activity, the average costs increased from EUR 0.4 to 9.26.

Inferential statistical analysis was conducted to compare the waste collection and disposal costs compared to the activation of door-to-door service with the municipal waste collection centers.

The distribution of these costs was not normal. (Shapiro–Wilk W-test = 0.78685, *p*-value < 0.0001). The nonparametric Kruskal–Wallis test was used, highlighting a difference between groups (Kruskal–Wallis chi-squared = 194.36, df = 3, *p*-value < 0.0001). Dunn’s post-hoc test subsequently highlighted no statistically significant difference between per capita costs of municipalities with a municipal waste collection center (MWCC) and without any service (*p* = 0.089) but costs were higher in municipalities with both door-to-door service and a municipal waste collection center (MWCC+DtD) compared to the municipalities with only a municipal waste collection center (MWCC) or to the municipalities with neither of two services (both *p* < 0.0001), and between those who have both door-to-door service and a municipal waste collection center (MWCC+DtD) and the municipalities with only door-to-door service (DtD) (*p* = 0.0057) (Figure 13).

## 7. Discussion

Numerous studies have tried to identify the factors influencing MSW production, separate collection, and management costs. Most of the studies on MSW management in Europe are concentrated on the cities areas and used every type of methods to predict MSW generation. MSW generation was predicted using regression and trend analysis in Romania [25] for example. A regression model was also used in Turkey [13]. An ANN model was used for example in Serbia [26]. Other studies developed a general regression neural network (GRNN) model for the prediction of annual municipal solid waste (MSW) generation at the national level for 44 countries of different size, population and economic development level [9]. Deep learning model combined with permutation-based VI scores method (R package “vip”) [44] and individual conditional expectation (ICE) plots method [45] was the very innovative and modern approach of the study of this paper to forecast MSW independent variables, to determine the factors that most influence the variables and to estimate how much they affect them. In addition, MSW production, separate collection, and MSW management costs with the same database were considered in this study. No literature paper was found considering these three variables at the same time. Deep learning models have proven very effective in predicting all three variables, with correlation between the real values in the testing dataset (30% of the data) and those predicted from the model between 86.2% and 94.8%. Statistical inference tests were used to reinforce the results of deep learning models on some important influencing factors.

In particular, many studies have shown that the factors are specific to the area of study. Each area has different local conditions such as climate, lifestyle, technological aspects, economy, and culture [58].

We highlighted that the type of residential structures is an important influencing factor for all three independents variables (production of waste, MSW separate collection and management costs). In particular, where the number of rooms in a residential building is low and medium (up to eight rooms), waste generation tends to be lower. If this number is exceeded, production of waste tends to be higher. These findings are consistent with some studies that have reported a directly proportional relationship between the size of buildings and the generation of waste [26,27,28,29,30,31,32,33,34,35,36,37,38,39,40,41,42,43,44,45,46,47,48,49,50,51,52,53,54,55,56,57,58,59].

In municipalities with a high percentage of larger buildings (larger than 16 rooms and with more than four floors), separate waste collection tends to be lower. This factor requires more in-depth studies. There can be many possible explanations for this finding: difficulties in organizing an efficient collection system and/or aspects more related to the prevailing socio-economic characteristics of the inhabitants of such buildings.

A further element to be explored is the importance of age factors in residential buildings. Where there are many older buildings, production of waste and costs for transport and disposal of waste tend to be lower. One of the possible explanations is linked to the abandonment rate of buildings.

Consistent with many studies [13,26,27,58,59,60] an important factor influencing MSW production is the annual income of the population (in particular, low- and medium-income classes up to EUR 55,000 per year). The environmental Kuznets curve theorized that at an initial increase in pollution (in this case the production of waste) is linked to increases in per capita income, but the curve has a climax, thereafter turning downward due to a greater willingness to pay for having a higher environmental quality [61]. In the Lombardia region, a study showed that the climax of the curve is between EUR 23,500 and 28,000 [62]. In Apulia region, we found production of waste increased as income class increase from EUR 0–10,000 (+7% of production for each percentage increase in this income range), EUR 10,000–15,000 (+7.8%), and EUR 15,000–26,000 (+10.5%), and then the curve began to flattening from EUR 26,000–55.000 (+6.8%) upward.

Some studies reported that in urban and tourist areas, waste production is greater than in rural areas [63]. The same finding is here reported for Apulia, where higher MSW amount and management costs were higher in coastal municipalities. In Apulia, the largest and tourist attraction urban centers are coastal. Tourist arrivals had an important impact both on separate MSW collection (among the top 20 influential factors) and especially on the costs of the collection and disposal of waste (directly proportional trend with this variable).

As expected, the factors that most influenced the separate collection were door-to-door services [64] and the collection of organic fraction. In particular, a high organic fraction had a strong impact on separate MSW collection due to the characteristics of MSW in Apulia. Consistent with other parts of the world [59,65,66], the organic fraction is one of the main product fractions of MSW (about 19% in Apulia per 2018 Waste Observatory data).

Municipal collection centers were found to have an important impact on increasing the percentage of separate collection, but did not significantly affect the individual costs of waste collection and disposal. This cost became significant when associated with door-to-door service.

## 8. Conclusions

We wanted to approach the problem of the production, separate collection, and management costs of MSW developing a multivariate analysis based on a new type of predictive model (deep learning approach) and considering a large number of factors (102 factors including, for example, infrastructure, organizational, demographic, social, and economic factors including organizational and economic services of treatment, waste disposal, and separate waste collection).

Deep learning model combined with permutation-based VI scores method (R package “vip”) and individual conditional expectation (ICE) plots method was the very innovative and modern approach of the study of this paper to forecast MSW independent variables, to determine the factors that most influence the variables and to estimate how much they affect them. This approach has proved particularly effective and reproducible in other territory.

Knowledge of the factors affect the three variables and how much they affect is an extremely valuable element to understand what would happen if you activate incident actions on these influencing factors (for example, we demonstrated the importance of combining services such as municipal collection centers and door-to-door waste collection service. This combination significantly increases separate collection, having associated benefits for both environmental and health. We can estimate how much differentiated collection could be increased and how much management costs could be increased, applying our model). So, many possibilities for cost-benefit analyses to evaluate policy to plan more effective and efficient MSW management services, to reduce waste generation, to increase MSW separate collection or to optimisation MSW management costs could be developed.

Future developments could refine these models and increase the accuracy of the resulting information. For example our model had a lack of information and data about educational qualifications and unemployment rates of the population.

## Figures and Tables

**Figure 1 ijerph-18-00752-f001:**
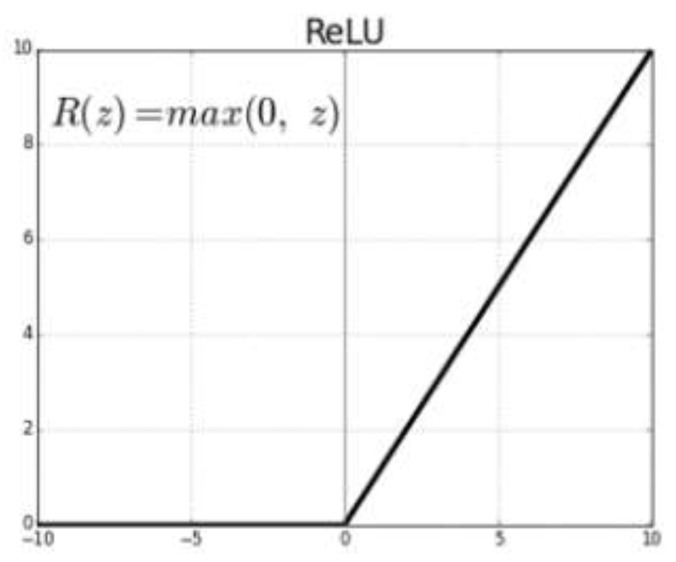
Rectified linear input (ReLU) activation function [33,34].

**Figure 2 ijerph-18-00752-f002:**
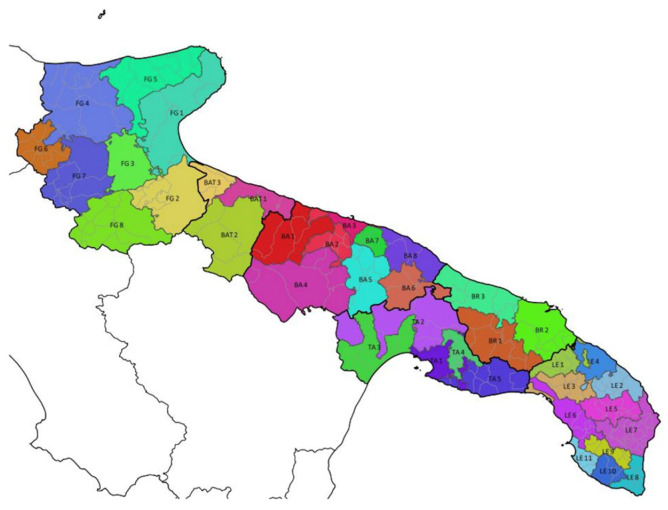
Homogenous areas of municipal solid waste (MSW) management services in Apulia [47].

**Figure 3 ijerph-18-00752-f003:**
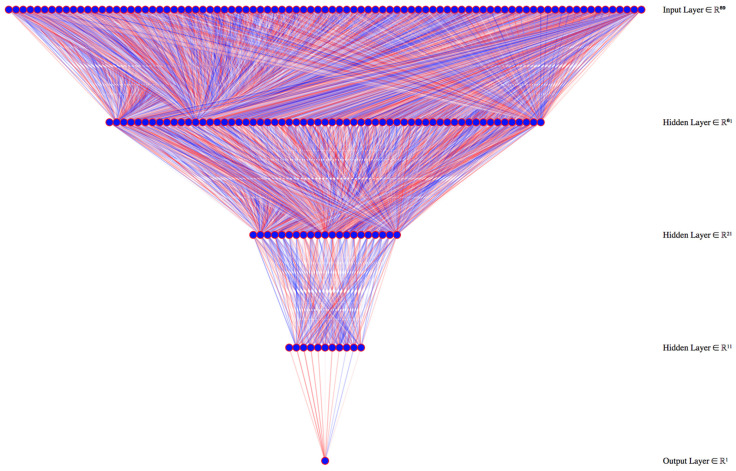
Deep learning model architecture for annual per capita waste production. We used 800 learning epochs were used to train the model.

**Figure 4 ijerph-18-00752-f004:**
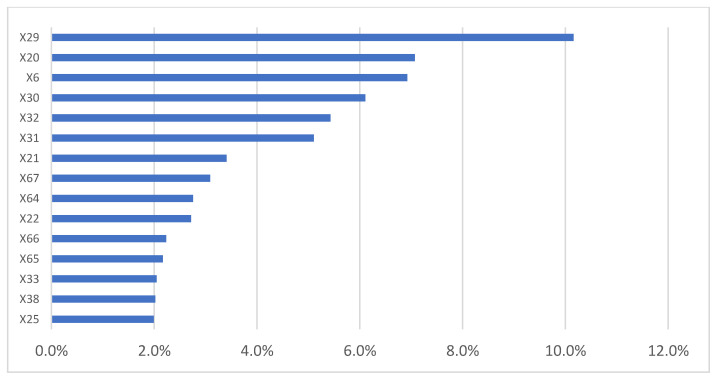
**The** 15 independent variables most influencing the annual per capita waste production (variable = X4).

**Figure 5 ijerph-18-00752-f005:**
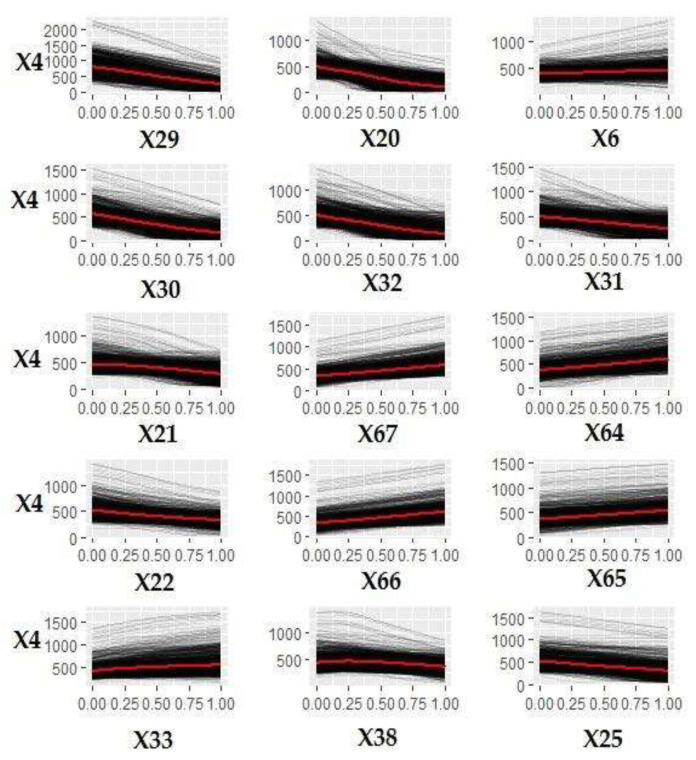
ICE curves of the 15 most influential variables on annual per capita waste production.

**Figure 6 ijerph-18-00752-f006:**
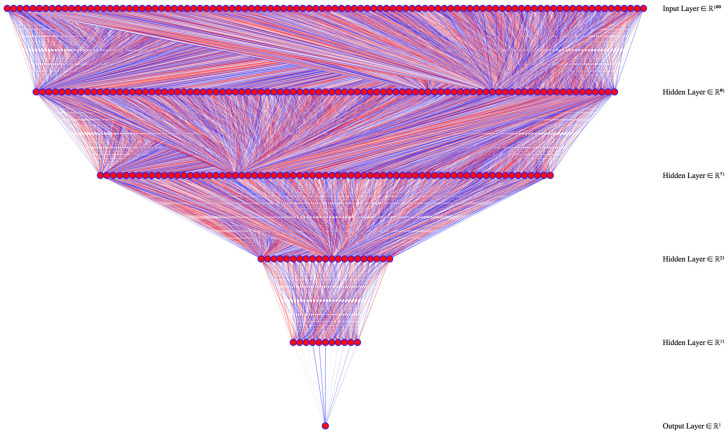
Deep learning model architecture for the percentage of separate collection, using 100 learning epochs to train the model.

**Figure 7 ijerph-18-00752-f007:**
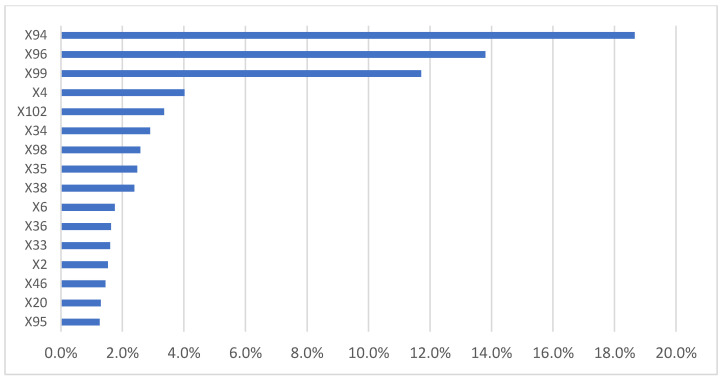
The 16 independent variables most influencing the percentage of separate collection (variable = X3).

**Figure 8 ijerph-18-00752-f008:**
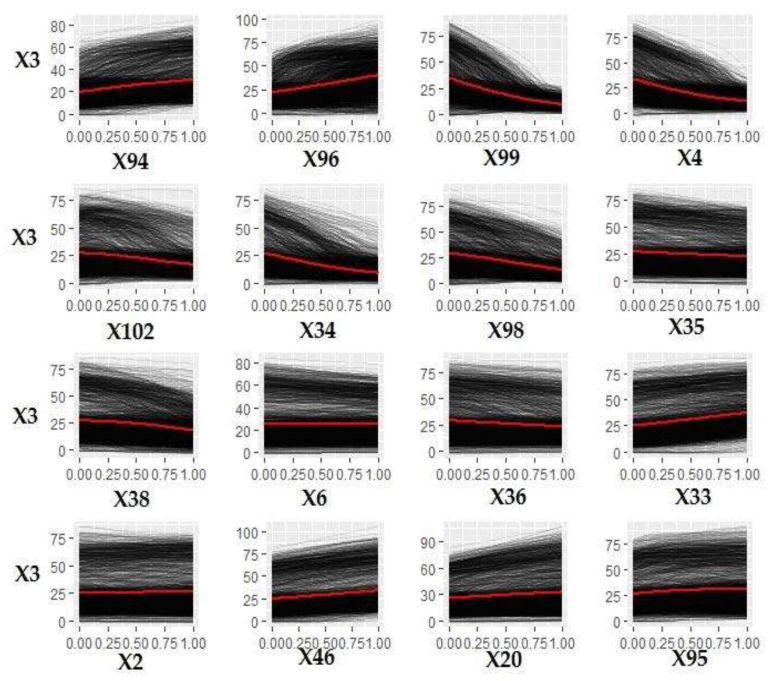
ICE curves of the 16 variables most influencing percentage of separate waste collection (X3)*; *X20 and X95 have the same percentage of influence.

**Figure 9 ijerph-18-00752-f009:**
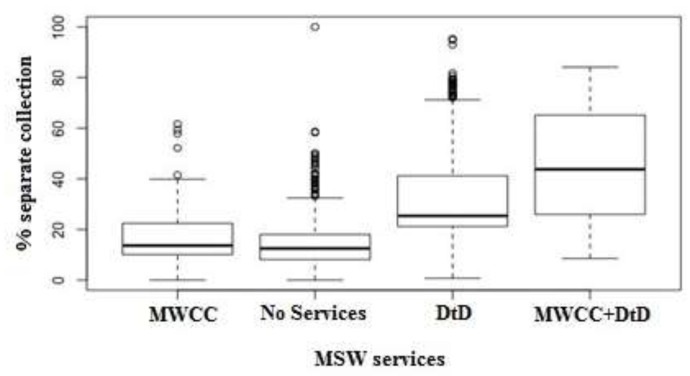
Boxplot percentage of separate collection comparing the presence of door-to-door service or municipal waste collection centers.

**Figure 10 ijerph-18-00752-f010:**
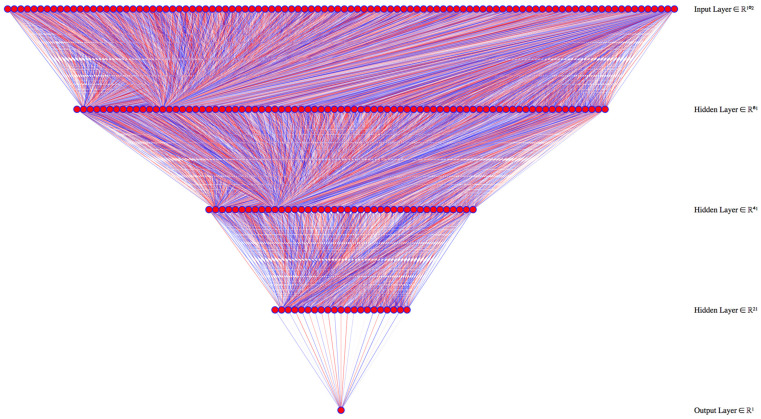
Deep learning model architecture for waste collection and disposal costs. We used 200 learning epochs to train the model.

**Figure 11 ijerph-18-00752-f011:**
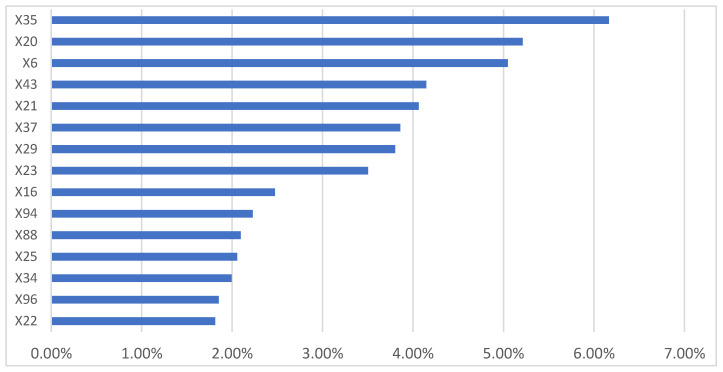
The 15 independent variables most influencing waste collection and disposal costs (variable = X1).

**Figure 12 ijerph-18-00752-f012:**
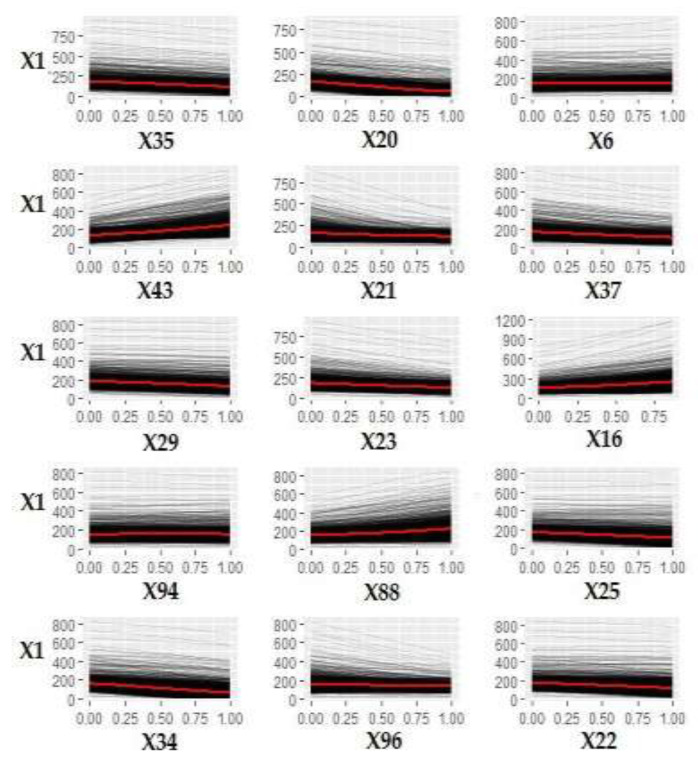
ICE curves of the 15 variables most influencing MSW management costs (X1).

**Figure 13 ijerph-18-00752-f013:**
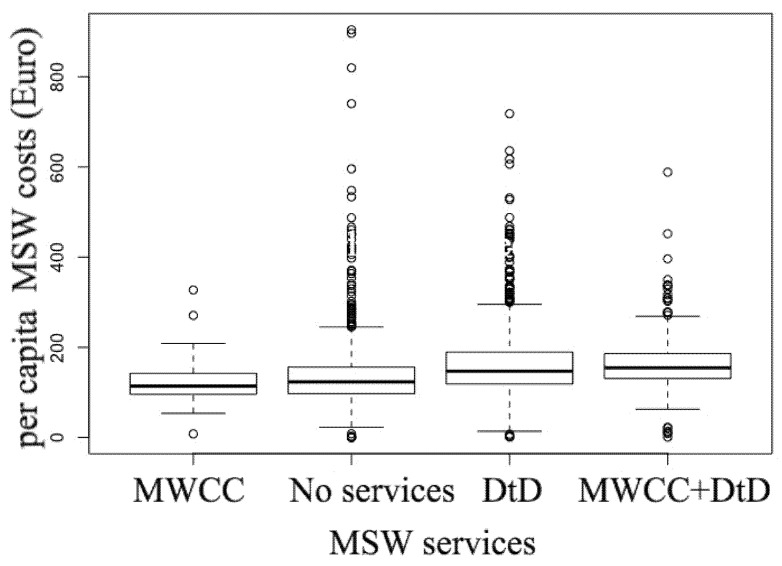
Boxplot of waste collection and disposal costs in comparison with the provision of door-to-door service or municipal waste collection centers.

**Table 1 ijerph-18-00752-t001:** Annual per capita waste production: Range of the average increase/decrease per unit of measurement of each of the 15 variables shown in Figure 5.

Cod.Var.	Variable	Measurement Unit	Maximum Increase/Decrease in Annual Production MSW at Variation of 1 Unit of Measurement (kg)	Medium Increase/Decrease in Annual Production MSW at Variation of 1 Unit of Measurement (kg)	Minimum Increase/Decrease in Annual Production MSW at Variation of 1 Unit of Measurement (kg)
X29	% residential buildings with 1 room/Tot. residential buildings	%	−7.1	−6.2	−5.3
X20	% residential buildings built <1918/Tot. residential buildings	%	−6.0	−5.0	−4.0
X6	Coastal or Rural Municipalities	Coastal/Rural	+160.4	+80.0	−0.4
X30	% residential buildings with 2 rooms/Tot. residential buildings	%	−6.3	−5.2	−4.2
X32	% residential buildings with 5–8 rooms/Tot. residential buildings	%	−14.7	−12.1	−9.5
X31	% residential buildings with 3–4 rooms/Tot. residential buildings	%	−8.1	−6.0	−3.9
X21	% residential buildings built 1918–1949/Tot. residential buildings	%	−6.8	−4.9	−2.9
X67	Annual income EUR 26,000–55,000/Total Income	%	+9.1	+6.8	+4.7
X64	Annual income EUR 0–10,000/Total Income	%	+9.3	+7.0	+4.7
X22	% residential buildings built 1946–1960/Tot. residential buildings	%	−5.8	−3.8%	−1.9
X66	Annual income EUR 15,000-–26,000/Total Income	%	+13.3	+10.5	+7.7
X65	Annual income EUR 10,000–15,000/Total Income	%	+11.4	+7.8	+4.2
X33	% residential buildings with 9–15 rooms/Tot. residential buildings	%	+9.3	+5.2	+1.0
X38	% residential buildings >4 rooms/Tot. residential buildings	%	+4.2	+2.1	0.0
X25	% residential buildings built 1981–1990/Tot. residential buildings	%	−7.5	−5.7	−3.9

**Table 2 ijerph-18-00752-t002:** Percentage of separate collection: Range of the average increase/decrease per unit of measurement of each of the 16 variables shown in Figure 9.

Cod. Var.	Variable	Measurement Unit	Maximum Increase/Decrease in % of Separate Collection at Variation of 1 Unit of Measurement	Medium Increase/Decrease in % of Separate Collection at Variation of 1 Unit of Measurement	Minimum Increase/Decrease in % of Separate Collection at Variation of 1 Unit of Measurement
X94	Door to Door Service	0 = No 1 = Yes all year round	+3.2%	+11.0%	+18.8%
X96	% Organic Fraction/Total separate collection	%	+0.16%	+0.08%	+0.002%
X99	% Paper Fraction/Total separate collection	%	−0.33%	−0.25%	−0.17%
X4	Annual per capita waste production	Kg	−0.022%	−0.016%	−0.011%
X102	% Multimaterial Fraction/Total separate collection	%	−0.26%	−0.14%	−0.03%
X34	% residential buildings with >16 rooms/Tot. residential buildings	%	−1.2%	−0.83%	−0.46%
X98	% Glass fraction/Total separate collection	%	−0.29%	−0.20%	−0.11%
X35	% residential buildings with 1 floor/Tot. residential buildings	%	+0.05	−0.05%	−0.15%
X38	% residential buildings > 4 floors/Tot. residential buildings	%	−0.43%	−0.22%	−0.02%
X6	Coastal or rural municipalities	Coastal/Rural	+5.8	−2.0%	−9.8%
X36	% residential buildings with >2 floors/Tot. residential buildings	%	+0.08%	−0.08%	−0.23%
X33	% residential buildings with 9–15 rooms/Tot. residential buildings	%	+1.05%	+0.64%	+0.24%
X46	% Land consumption/total land extension	%	+0.39%	+0.20%	0.0%
X20	% residential buildings built <1918/Tot. residential buildings	%	+0.19%	+0.09%	0.0%
X95	% separate waste collection gathered by municipal waste collection center /total separate collection	%	+0.18%	+0.1	+0.02%

**Table 3 ijerph-18-00752-t003:** MSW management costs: Range of the average increase/decrease per unit of measurement of each of the 15 variables shown in Figure 13.

Cod.Var.	Variable	Measurement Unit	Maximum Increase/Decrease in Waste Collection and Disposal Costs at Variation of 1 Unit of Measurement (EUR)	Medium Increase/Decrease in Waste Collection and Disposal Costs at Variation of 1 Unit of Measurement (EUR)	Minimum Increase/Decrease in Waste Collection and Disposal Costs at Variation of 1 Unit of Measurement (EUR)
X35	% residential buildings with 1 floor/Tot. residential buildings	%	−1.72	−0.87	−0.02
X20	% residential buildings built <1918/Tot. residential buildings	%	−2.24	−1.43	−0.64
X6	Coastal or Rural Municipalities	Coastal/Rural	+74.2	+10.0	−54.2
X43	% divorced population/Tot. Inhabitants	%	+44.52	+27.6	+10.73
X21	% residential buildings built 1919–1945/Tot. residential buildings	%	−2.39	−0.84	+0.70
X37	% residential buildings with 3 floors/Tot. residential buildings	%	−2.60	−1.31	−0.01
X29	% residential buildings with 1 room/Tot. residential buildings	%	−1.40	−0.67	+0.05
X23	% residential buildings built 1961–1970/Tot. residential buildings	%	−3.63	−1.83	−0.02
X16	Tourist arrivals	N	+0.0003	+0.0002	+0.00005
X94	Door-to-door service	0 = No 1 = Yes all year round	+69.2	+5.0	−59.2
X88	I—Accommodation and catering services—local units of active enterprises per km^2^	N	+9.26	+4.82	+0.4
X25	% residential buildings built 1981–1990/Tot. residential buildings	%	−2.82	−1.36	+0.09
X34	% residential buildings >16 rooms/Tot. residential buildings	%	−7.78	−4.74	−1.70
X96	% Organic Fraction/Total separate collection	%	−1.04	−0.40	+0.24
X22	% residential buildings built 1946–1960/Tot. residential buildings	%	−2.97	−1.44	+0.58

## Data Availability

Data used in this paper are available in public accessible repositories available at the links indicated in Section 5.3.

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
