# Peer review of "Variables Influencing per Capita Production, Separate Collection, and Costs of Municipal Solid Waste in the Apulia Region (Italy): An Experience of Deep Learning"

_ijerph, 2021, doi:10.3390/ijerph18020752_

Round 1
Reviewer 1 Report
The paper gives a very intruiging insight into the municipal solid waste conundrum. The paper has a very strong dataset backed by novel models. It will be of great interest to the waste management field. You may want to edit te document for minor grammatical mistakes. Also consider changing the presentation and layout of the methodology section as it lacks a coherent flow. Otherwise a very good paper.
Author Response
Response to Reviewer 1 Comments
Dear Reviewer, thank you for your comments that helped us to improve our paper. Below are the answers to your comments
Point 1: The paper gives a very intruiging insight into the municipal solid waste conundrum. The paper has a very strong dataset backed by novel models. It will be of great interest to the waste management field. You may want to edit document for minor grammatical mistakes. Also consider changing the presentation and layout of the methodology section as it lacks a coherent flow
Response 1: We changed the presentation expanding the bibliographical references and trying to make the concepts more consistent and clear. Moreover we reorganize methodology section to make clearer and more coherent steps developed in the study. We also reviewed the text format and unify it (specially the underlined)
Kind regards

Reviewer 2 Report
Dear authors, I think that your paper has potentialities. My recommendation is accepting the article since the research activities are also quite interesting and meaningful and the topic investigated is within the boundaries of the journal. However, the structure should be improved, and some technical explanations should be also clarified.
INTRODUCTION: Paper states that many studies showed that many factors affect the MSW management systems but more background information needs to be included. I suggest supporting your paper with more references. What has been done in multivariate analysis of MSW management already for Europe? and for Italy? Maybe this could be provided in a different “background” section after the Introduction.
DEEP LEARNING THEORY AND INTEGRATED WASTE MANAGEMENT THEORY: Personally, I think that these stages may be reorganised. The authors should expect a close-to-complete rewrite and restructure these stages of the manuscript, improving presentation of a complete state-of-the-art section. Moreover, some potentially controversial affirmations need to be supported with some references in order to make them more consistence (e.g. the affirmation that the use of incineration to recover energy should be limited to the minimum).
STUDY DESING: Figure 2 is confusing. Please, delete the numbers or re-number them to facilitate the understanding of the figure (If a reader tries to understand the figure only from the caption without reading the text, the first step of the methodology cannot be numbered as 4.1.1). Moreover, the figure is not useful enough as it repeats the same steps for each question. I suggest to modify the figure and include specific information for last stage in order to make it more comprehensive.
Several terms are used but never defined in order to specify what it consists of or what activities include (e.g. "waste collection center", "waste management services", etc.). Which is the difference between door-to-door services and waste management services? All the terms need to be clearly defined and used consistently. Does the term "waste production" include only municipal waste or also urban waste, since it seems you include “green waste” (line 160) as a fraction of separate collection? Waste composition also depends on socio-economic characteristics of producers, so do you assume that the waste composition is constant in all region? You are missing the waste composition. Please, clarify it.
Personally, I think that some information is confusing. For example, Land consumption gathered data (line 165) need to be deeply explained. The same occurs with the MSW management costs (What expenses are included in these costs?)
Line 204: What are the characteristics of a homogeneous area? Please, clarify it.
Line 208: I think that information related to the management services is quite confusing. Which are the differences between municipal company or public company? Which are the differences between “service managed with a consortium” and “consortium management service (head or body)”? Please clarify it.
Line 219: Are you missing the waste collection frequency (daily/weekly/etc)? Please, clarify and justify it.
In addition, I suggest to report more data related to the analysed factors (data collected for independent factors). Otherwise, the results are not reliable if presented in this form. Moreover, more information about the inconsistent or untreatable data are needed in order to know what was excluded from the analysis.
CONCLUSIONS: The conclusions section is considerably shorter and does not provide sufficient level of detail on what has been concretely achieved. Conclusions needs to be based on the results and put them in perspective with the problem outlined in the introduction section.
Figure 10, 14: The acronyms are not explained in neither text or caption.
Line 148. Please, correct the sentence: “Demographic, social. and infrastructural data”
Line 188: please, include the QGIS 3.10 reference.
Line 190: please, correct the format used.
All the acronyms should be defined in the text the first time they appear (see line 159)
Please, review the text format and unify it (specially the underlined)
Author Response
Response to Reviewer 2 Comments
Dear Reviewer, thank you for your comments that helped us to improve our paper. Below are the answers to your comments
Point 1: INTRODUCTION: Paper states that many studies showed that many factors affect the MSW management systems but more background information needs to be included. I suggest supporting your paper with more references. What has been done in multivariate analysis of MSW management already for Europe? and for Italy? Maybe this could be provided in a different “background” section after the Introduction.
Response 1: We expanded the bibliographical references of the Introduction section and trying to make the concepts more consistent and clear. Moreover we introduced par. 2 “Background of Multivariate analysis on MSW management in Europe and in Italy” describing several studies about multivariate analysis of MSW management in Europe and in Italy
Point 2: DEEP LEARNING THEORY AND INTEGRATED WASTE MANAGEMENT THEORY: Personally, I think that these stages may be reorganized. The authors should expect a close-to-complete rewrite and restructure these stages of the manuscript, improving presentation of a complete state-of-the-art section. Moreover, some potentially controversial affirmations need to be supported with some references in order to make them more consistence (e.g. the affirmation that the use of incineration to recover energy should be limited to the minimum)
Response 2: We restructured and improving Deep Learning Theory section introducing story of Machine Learning and Deep Learning, a summary of principle models and of the principle advantage to use these models (par.3). Moreover we clarified some affirmation in section Integrated Waste Management Theory. This section reported an European Law compendium about Waste Management. So, the affirmation that use of incineration and unsorted landfills should be limited to the minimum necessary is declared by European Law. Anyway, we introduced in the line 182-185 some motivation found in literature that can justified this choice and that reinforce the concept.
Point 3: STUDY DESING: Figure 2 is confusing. Please, delete the numbers or re-number them to facilitate the understanding of the figure (If a reader tries to understand the figure only from the caption without reading the text, the first step of the methodology cannot be numbered as 4.1.1). Moreover, the figure is not useful enough as it repeats the same steps for each question. I suggest to modify the figure and include specific information for last stage in order to make it more comprehensive.
Response 3: We decided to delete figure 2 and to reorganize the Section Study Design. We hope that this reorganization make clearer steps followed in study design
Point 4: Several terms are used but never defined in order to specify what it consists of or what activities include (e.g. "waste collection center", "waste management services", etc.). Which is the difference between door-to-door services and waste management services? All the terms need to be clearly defined and used consistently. Does the term "waste production" include only municipal waste or also urban waste, since it seems you include “green waste” (line 160) as a fraction of separate collection? Waste composition also depends on socio-economic characteristics of producers, so do you assume that the waste composition is constant in all region? You are missing the waste composition. Please, clarify it
Response 4: In the new paragraph 5.3 for each data collected we introduced some sentences to explain several terms used (e.g. from line 315 to line 320 are explained the terms of waste collection center and of door-to-door services). We explained in the text (line 290-293) that our study consider MSW production data included waste type consisting of everyday items that are discarded by the public. Although the waste may originate from a number of sources that has nothing to do with a municipality, the traditional role of municipalities in collecting and managing these kinds of waste have produced the particular etymology “municipal”. We do not assume that the waste composition is constant in all region. We specified in the paper (line 191-194) that Annual data from 2008 to 2018 on each municipalities were collected. In this way, the analyses included the different socio-economic characteristics of producers and their evolution.
Point 5: Personally, I think that some information is confusing. For example, Land consumption gathered data (line 165) need to be deeply explained. The same occurs with the MSW management costs (What expenses are included in these costs?)
Response 5: We explained better what is Land consumption (lines 303-306) and What expenses are included in MSW management costs (lines 309-310)
Point 6: Line 204: What are the characteristics of a homogeneous area? Please, clarify it.
Response 6: We introduced a sentence (lines 271-274) about it.
Point 7: Line 208: I think that information related to the management services is quite confusing. Which are the differences between municipal company or public company? Which are the differences between “service managed with a consortium” and “consortium management service (head or body)”? Please clarify it.
Response 7: We explain each term from line 352 to line 367
Point 8: Line 219: Are you missing the waste collection frequency (daily/weekly/etc)? Please, clarify and justify it.
Response 8: We clarified Annual data from 2008 to 2018 on each municipalities were collected. We also had monthly data about MSW, but data about influencing factor on MSW considered in our study, were really few at the municipal and monthly level.
Point 9: In addition, I suggest to report more data related to the analysed factors (data collected for independent factors). Otherwise, the results are not reliable if presented in this form. Moreover, more information about the inconsistent or untreatable data are needed in order to know what was excluded from the analysis.
Response 9: We introduced more information about the inconsistent or untreatable data in order to know what was excluded from the analysis form line 373 to line 377. We can attached list of 102 factors used in our study with the details of which of these factors were used for each independent variables (Annual per capita waste production, MSW separate collection and MSW management costs)
Point 10: CONCLUSIONS: The conclusions section is considerably shorter and does not provide sufficient level of detail on what has been concretely achieved. Conclusions needs to be based on the results and put them in perspective with the problem outlined in the introduction section.
Response 10: We improved discussion and conclusion paragraphs, giving greater emphasis to the good results derived from the deep learning models developed, in accordance with the problem outlined in the introduction section.
Point 11: Figure 10, 14: The acronyms are not explained in neither text or caption.
Response 11: See lines 481-485 and lines 534-539
Point 12: Line 148. Please, correct the sentence: “Demographic, social. and infrastructural data”
Response 12: See line 283
Point 13: Line 188: please, include the QGIS 3.10 reference.
Response 13: See lines 332-333
Point 14: Line 190: please, correct the format used.
Response 14: See lines 334
Point 15: All the acronyms should be defined in the text the first time they appear (see line 159)
Response 15: See lines 297
Point 16: Please, review the text format and unify it (specially the underlined)
Response 16: We reviewed the text format
King Regards

Reviewer 3 Report
Dear authors:
I think you did an interesting work, but the manuscript need some improvement before publication.
Find some comments below:
- Lines 30 to 129 should not be underlined
- More updated references about this topic should be added as references given are older than 5 years ago. Some examples: Solano Meza, J.K.; Orjuela Yepes, D.; Rodrigo-Ilarri, J.; Cassiraga, E. Predictive Analysis of Urban Waste Generation for the City of Bogotá, Colombia, through the Implementation of Decision Trees-Based Machine Learning, Support Vector Machines and Artificial Neural Networks. Heliyon 2019, 5, e02810.
- Materials and methods are not properly explained. The methodology shown in Figure 2 should be explained in detail before referring to the case study in Apulia.
- In other words, the case study should be just an illustration of the proposed methodology and you should not use the case study to develop the methodology so it is somehow "self-explained".
- Discussion section should be expanded, taking into account results shown before in sections 5.1, 5.2 and 5.3. Comparisons between them should be made, so conclusions are supported by these results.
- Conclusions should be rewritten and expanded too. They are too short and vague.
Author Response
Response to Reviewer 3 Comments
Dear Reviewer, thank you for your comments that helped us to improve our paper. Below are the answers to your comments
Point 1: Lines 30 to 129 should not be underlined
Response 1: We reviewed the text format deleted the underlined.
Point 2: More updated references about this topic should be added as references given are older than 5 years ago. Some examples: Solano Meza, J.K.; Orjuela Yepes, D.; Rodrigo-Ilarri, J.; Cassiraga, E. Predictive Analysis of Urban Waste Generation for the City of Bogotá, Colombia, through the Implementation of Decision Trees-Based Machine Learning, Support Vector Machines and Artificial Neural Networks. Heliyon 2019, 5, e02810.
Response 2: We expanded the bibliographical references of the Introduction section with more recent references (including also the article kindly suggested)
Point 3: Materials and methods are not properly explained. The methodology shown in Figure 2 should be explained in detail before referring to the case study in Apulia. In other words, the case study should be just an illustration of the proposed methodology and you should not use the case study to develop the methodology so it is somehow "self-explained".
Response 3: We decided to delete figure 2 and to reorganize Materials and methods like you suggested . We hope that this reorganization make clearer steps followed in study design
Point 4: Discussion section should be expanded, taking into account results shown before in sections 5.1, 5.2 and 5.3. Comparisons between them should be made, so conclusions are supported by these results. Conclusions should be rewritten and expanded too. They are too short and vague.
Response 4: We improved discussion and conclusion paragraphs, giving greater emphasis to the good results derived from the deep learning models developed, in accordance with the problem outlined in the introduction section.
Kind Regards

Round 2
Reviewer 3 Report
Dear Authors
your paper has been updated as suggested and I think it is now ready for publication.
This manuscript is a resubmission of an earlier submission. The following is a list of the peer review reports and author responses from that submission.
Round 1
Reviewer 1 Report
The paper is a very novel piece giving a new methodological approach to Municipal Solid Waste and the authors are commended for that. You need to consider the following issues to enhance your publication to an acceptable standard:
- The document needs some major language editing. There are a lot of errors that need to be fixed.
- The paper could be enhanced by the addition of Waste management related theory before the materials and methods section. This will give readers a good understanding into the methods used warranted by such theory.
- There are inconsistencies in the layout of your section 2. The document will read better if the formatting was standardized.
- The results seem to jump out at the readers. a brief introduction to this section will help ease readers into this section.
Reviewer 2 Report
I would expect a clear motivation for the study in the introductory passage. The rationale for the study is obscure and difficult to relate with.
Authors need to place the study in context of previous studies and state how current study improves existing knowledge.
style of writing is a bit everywhere. Authors need to compress the ideas being expressed in the study. In fact, it is challenging to comprehend the flow of ideas.
The method section should be described in the context of the research gap. That is, how the method is appropriate to resolve the problem identified.
The results' presentation needs improvement because, it needs to be clear how the findings differ or support previous studies. However, the discussion is well presented.
Essentially, this document will benefit from the services of an English Language editor.
Reviewer 3 Report
Waste generation, collection, and the associated costs are all important topics, in the realm of waste management and it makes very much sense to bring cross-disciplinary ideas to nurture the topics, as suggested by the title of this paper. Therefore, I started reading this manuscript with interest, only to be disappointed in the middle.
This is a poorly constructed manuscript on a "potentially" interesting topic. Authors have not made a strong case to justify why the research has been carried out, and also they have not provided the scope and objectives of the research in a scientific way. Sections 1 and 2 have not been constructed properly and remind a chopped-down version of a project report. Although, the next sections (3 and 4) are bit better in presentation, it is not easy to follow, due to the poor usage of English, which seemed to be an obvious issue throughout the paper.
I am unable to recommend this paper to be published in any scientific journal.